behaviour/ecology

rhythm analysis, acoustic communication, fish sounds, PAM, periodicity, animal communication

**Author for correspondence:**
Lara S. Burchardt
e-mail: lara.burchardt@mfn.berlin

# A primer on rhythm quantification for fish sounds: a Mediterranean case study

Lara S. Burchardt[1,2], Marta Picciulin[3], Eric Parmentier[4] and Marta Bolgan[4]

[1]Museum für Naturkunde – Leibniz Institute for Evolution and Biodiversity Science, Invalidenstraße 43, 10115 Berlin, Germany
[2]Institute of Animal Behaviour, Freie Universität Berlin, Takustr. 6, 14195 Berlin, Germany
[3]Ca' Foscari University of Venice, Italy
[4]Laboratory of Functional and Evolutionary Morphology (Freshwater and Oceanic sCience Unit of reSearch), Institut de Chimie B6c, University of Liège, Liège, Belgium

(iD) LSB, 0000-0002-9210-7934

We have used a lately established workflow to quantify rhythms of three fish sound types recorded in different areas of the Mediterranean Sea. So far, the temporal structure of fish sound sequences has only been described qualitatively. Here, we propose a standardized approach to quantify them, opening the path for assessment and comparison of an often underestimated but potentially critical aspect of fish sounds. Our approach is based on the analysis of inter-onset-intervals (IOIs), the intervals between the start of one sound element and the next. We calculate exact beats of a sequence using Fourier analysis and IOI analysis. Furthermore, we report on important parameters describing the variability in timing within a given sound sequence. Datasets were chosen to depict different possible rhythmic properties: *Sciaena umbra* sounds have a simple isochronous—metronome-like— rhythm. The /Kwa/ sound type emitted by *Scorpaena spp.* has a more complex rhythm, still presenting an underlying isochronous pattern. Calls of *Ophidion rochei* males present no rhythm, but a random temporal succession of sounds. This approach holds great potential for shedding light on important aspects of fish bioacoustics. Applications span from the characterization of specific behaviours to the potential discrimination of yet not distinguishable species.

## 1. Background

**Rhythms** (i.e. non-random, ordered and recurrent alternation of different **elements** in a **sequence** of sounds and silence) can be found everywhere in the world, from human speech to music and animals' communication [1,2]. They can be an inherent

property of sound production when coupled to other physiological processes such as breathing or wingbeat [3–6]. Rhythms can code for individual or context-related information. They are for example used to discriminate between familiar and unfamiliar opponents in northern elephant seals [7]. They encode urgency in chickadees (*Poecileatricapillus)*, as well as in different species of ground squirrels, while distinct call categories with different temporal structures—i.e. rhythms—code for different predators in meerkats (*Suricata suricatta*) [8–10]. Some birds use well-known musical tempo markings to attract listeners, such as accelerando or ritardando (speeding up and slowing down, respectively) [11]. An advantage of producing rhythmic patterns rather than random patterns is that listeners might better anticipate rhythmic patterns. The listeners are therefore less likely to miss the next element of an **acoustic signal** sequence. This is only true if the respective species are capable of perceiving the rhythmic pattern. That was discussed for example for male zebra finches' or bats' **isochronous** (i.e. metronome-like beats with similar intervals between elements) acoustic signals [12,13]. For zebra finches, we know that the sequential dependencies of isochronous patterns can be perceived across many tempi [14]. Not all rhythmic patterns need to be isochronous though; a more complex rhythm with more than one underlying **beat** would be called **heterochronous** [2,15]. Tuning attention to the perceived rhythm can reduce 'attentional energy' (*sensu* [16]) and increase signal perception [13], again assuming the ability to perceive the respective patterns. Important terminology with regards to rhythm, used in this study, is explained in table 1, and the terms that are explained there are written in bold on the first appearance in the text.

In many fish species, sound production is an important communication modality that reflects several vital biological processes (such as spawning, courtship, feeding, social cohesion and competition), and that promotes mate choice and mating success [17,18]. Many vocal fish produce highly stereotyped, pulsatile sounds, generated by rhythmic contractions of muscles, acting under a rhythmically active vocal motor network [19]. As such, the **periodicity** *within* the fish sound elements can be indicative of the periodical activity of neurophysiological processes [20,21]. The measurement and analysis of temporal features *within* single fish sound elements are routinely included in fish bioacoustics studies. Those *within-sound* temporal features can code for species identity [22] as well as for specific behavioural patterns or motivational states [23–25]. Furthermore, *within-sound* temporal features can be influenced by environmental conditions such as water temperature, as this can impact the performance of the morpho-physiological processes underlying sound production [26,27]. It can also be affected by man-made acoustic disturbances [28]. If periodicity can be found *within* fish sounds, this could also be expected to occur, to some extent, *between* fish sounds. In this sense, we will analyse the temporal structure *between* pulsatile and non-pulsatile fish sounds, which will be referred to as an 'element' (figure 1), following Burchardt & Knörnschild [2]. In our case, an element is the entirety of one pulsatile sound surrounded by silence (figure 1a). A sequence is composed of adjacent elements that are regarded as being produced by the same individual, representing a unit in terms of context or timing. A sequence can be terminated if it is disturbed by a different individual or after a prolonged silence. Sequences can range from only a few elements to containing several hundred elements, as is the case in bird song or long echolocation sequences in sperm whales [2,29].

In fish, a reliable methodology for the quantification of the temporal periodicity along a sequence of elements (i.e. rhythms), namely the periodicity *between* elements, is currently lacking. Quantifying fish rhythms holds great potential to shed light on overlooked, potentially biologically significant aspects of fish bioacoustics. Using a recently developed workflow, and the corresponding instructions for the quantification of rhythms in acoustic signals optimized on other taxa [12,13], we aim to fill this gap.

The methods proposed here were previously developed and tested, among others, on element sequences of zebra finches and bats, which are both characterized by isochronous beats [2,12,13]. These approaches are based on the analysis of **inter-onset-intervals** (IOIs), the intervals between the start of one sound element and the start of the next. Important parameters describing the variability in the timing of a given sound sequence are furthermore analysed. Finally, this methodology permits to assess the exact beat of a sequence, as calculated by two different methods: Fourier analysis [2,15,30], where any continuous signal is decomposed into its sinusoidal components, which are nothing but frequencies (Hz; as in beats per second), and IOI calculations, using the mean of IOIs to calculate a frequency [2,12,15]. A best-fitting beat is the one beat in hertz (as in beats per second) that best describes (i.e. fits) the temporal structure of elements in a sequence.

Three fish sound types recorded in different areas of the Mediterranean Sea were used to test the methodology. The first sound type is emitted by the brown meagre *Sciaena umbra* Linnaeus 1758 (Perciformes). This species has been reported to produce three reproductive vocal patterns in the wild: the so-called R-pattern (figure 1a), qualitatively described as showing a rhythmic structure between

**Table 1.** Glossary of rhythm terminology (adapted from Burchardt and Knörnschild [2]).

| acoustic signals | all acoustic signals that animals produce voluntarily |
| --- | --- |
| rhythm | e.g. non-random, ordered, and recurrent alternation of different elements in a sequence of sounds and silence in speech, music or animals' acoustic signals |
| periodicity | an underlying reoccurring pattern describing a sequence as periodic, e.g. an isochronous pattern |
| isochrony | a metronome-like beat with the same intervals between elements, very simple; in contrast with heterochrony |
| heterochrony | a pattern with more than one underlying beat |
| beat/beat frequency | the unit to describe a pattern, given in hertz (beats per second); a beat frequency of 5 Hz would describe a sequence with an underlying pattern of five beats per second, i.e. five elements per second or a temporal structure where elements are distributed regularly in a way that you could fit a maximum of five elements into 1 s |
| inter-onset-interval (IOI) | in a sequence of acoustic signals, the period between the start of an element and the next element, comprising the element duration and the following gap duration; in the case of fish, this refers to the interval between sounds; in other taxa or contexts also called inter-pulse-interval, inter-click-interval or inter-call-interval |
| element | the subunit of a sequence of acoustic signals, i.e. a distinct sound, syllable, call, click, etc. surrounded by silence, IOIs are measured between elements; in fish, this corresponds to the sound |
| sequence | several adjacent elements uttered by the same individual, i.e. the analysis unit of the rhythm analysis; a beat is calculated per sequence |
| normalized pairwise variability index (nPVI) | a measure of the average variation of a set of IOIs, which describes how well the next IOI can be predicted based on the previous IOI; the nPVI would be 0 in a perfectly isochronous sequence; higher nPVI values correspond to higher variability between IOIs |
| coefficient of variation | a parameter describing the variability of a dataset; it is the standard deviation set into relation with the sample mean of the IOIs within a sequence; independent of sample size and mean, and therefore providing a comparable description of a sequence's variability |

elements; an irregular pattern (I-sound calling pattern) which lacks visible, fixed repetition of elements; and the chorus (i.e. mass production of sounds in which individual elements are not detectable) [25]. We used the R-pattern (figure 1*a*) in this study to validate the existence of a rhythmic pattern as it is suggested by the qualitative analysis of spectrograms.

The second sound type is called /Kwa/ and is most probably produced by species from the ray-finned genus *Scorpaena* (Scorpaeniformes). The qualitative inspection of /Kwa/ sequences recorded in the wild suggests a non-random patterning (figure 1*b*) between elements, but the single observation does not allow the association of an isochronous rhythm [31]. Therefore, this sound type was chosen to quantitatively ascertain the presence of periodicity, in a case in which qualitative inspections of the audio tracks do not provide a conclusive answer.

The third sound type was chosen to represent a case in which regularity between elements cannot be observed at a qualitative level, i.e. the calls emitted by Roche's snake blenny, *Ophidion rochei*, males Müller 1845 (Ophidiiformes) [32]. In this case, the elements are produced with long, irregular silent gaps (figure 1*c*), and it therefore does not necessarily fulfil our sequence definition, after which a sequence is terminated after a defined species-specific duration of silence. Nevertheless, we chose these sounds to provide an example where no rhythm is present.

We aim to provide an overview of the performances and interpretation of this methodology when applied to different fish sounds recorded at sea. To that end, we tested it on the three described fish sound types, i.e. (i) a case in which qualitative inspection of acoustic tracks suggests the presence of

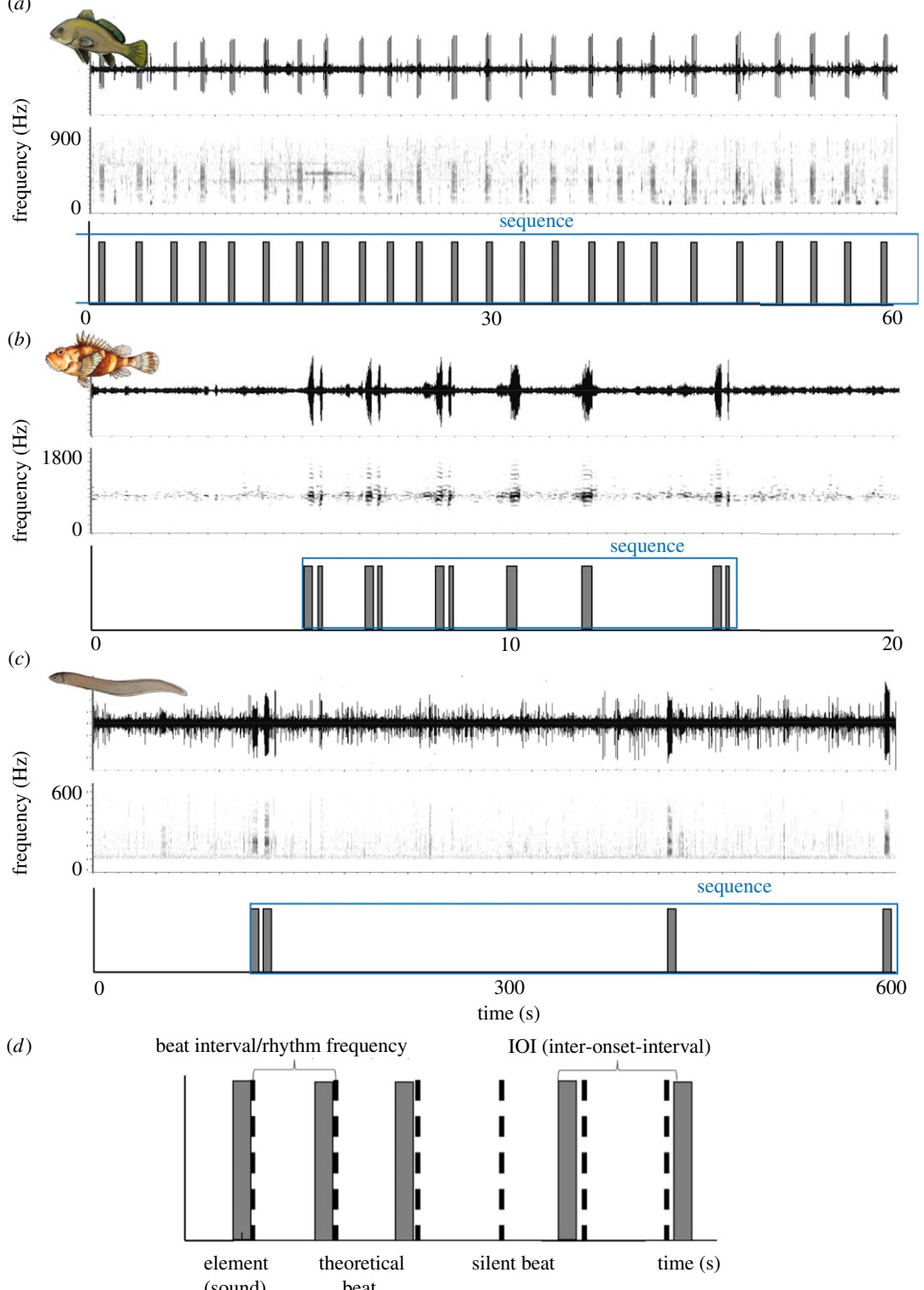

**Figure 1.** Waveform, spectrogram (Hanning, 256, 50% overlap) and bar plot of representative (*a*) *Sciaena umbra*, (*b*) /Kwa/ and (*c*) *Ophidion rochei* acoustic samples analysed in this manuscript. Each bar in the bar plots corresponds to an element (i.e. sound); bar width corresponds to element duration. Note the different time scales of the three panels, as well as the degree of variability in number and duration of the elements between taxa; (*d*) schematic depicting the adopted terminology.

regular periodicity between elements (figure 1*a*), (ii) a case in which a more complex periodicity is suggested by the qualitative analysis of spectrograms (figure 1*b*), and (iii) a case in which visual inspection of the spectrograms does not suggest a regular periodicity between elements (figure 1*c*). The three sound types presented here overall provide good examples of the qualitative variability

(independent of rhythmicity) which can be expected between fish sound sequences emitted by different taxa. This variability can involve (i) sequence duration, which can range from a few seconds (e.g. /Kwa/ sound type) to several minutes (e.g. *S. umbra*); (ii) the number of elements within a sequence, which can range from few elements (e.g. /Kwa/ sound type) to hundreds of elements (e.g. *S. umbra*); and (iii) the duration of the elements, which can span from a few milliseconds (e.g. /Kwa/ sound type) to seconds (e.g. *O. rochei* sounds) (figure 1). This variability is likely to be expected between other fish sound types and taxa and therefore it is necessary to establish and evaluate the performance of a quantitative methodology accounting for this variability.

# 2. Methods

## 2.1. Sound recordings and labelling of elements

Rhythm analysis was conducted on element sequences emitted by species or genus of three different orders, namely *Sciaena umbra*, *Ophidion rochei* and *Scorpaena spp*.

*Sciaena umbra* sound files containing the R-sound pattern were recorded during summer 2019 along the Venice inlets (Italy; 45.4130°N, 12.2972°E) by using a pre-amplified GP1280 hydrophone (Colmar SRL, La Spezia, IT: sensitivity, −170 dB re. 1 V Pa$^{-1}$; frequency range, 5 Hz–90 kHz) connected to a Tascam Handy Recorder (Tascam, Montebello, CA) generating WAV files at 44.1 kHz and 16-bit depth [25].

*Sciaena umbra*'s R-pattern was also recorded in the Gulf of Trieste (Italy) [33]. Here, recordings including both *S. umbra* sounds and *Ophidion rochei* [32] sounds were collected during summer 2009 in the core zone of the Miramare Marine Protected Area (45.7018°N, 13.7126°E) by using an acoustic data logger prototype provided with a pre-amplified Reson TC 4013 hydrophone (sensitivity −170 dB re 1 V μPa$^{-1}$, frequency range 1 Hz to 170 kHz) and a Gemini iKey Plus Recording Device (10 min wav files, rate of 44.1 kHz, 16 bits).

Finally, acoustic recordings including the /Kwa/ sound type were collected during July 2017 in *Posidonia oceanica* meadows located at −20 m depth in Palma Bay Marine Reserve (Mallorca, Spain: 39.6958°N, 2.7258°E) using an underwater acoustic data logger SNAP (Loggerhead Instruments, FL, USA: hydrophone sensitivity; −170 dB re 1 V μPa$^{-1}$), recording WAV files of 1 min every 5 min (over the 24 h) at 44.1 kHz, 16 bits.

Fish sound sequences were defined as the temporally ordered repetition of at least three elements (the entirety of one pulsatile sound surrounded by silence), which are temporally and clearly isolated from preceding and following elements. In data collected at sea, a sequence can be confidently assigned to the same individual if all elements within the sequence are characterized by the same signal-to-noise ratio. This method was also applied here. We are, therefore, confident that one sequence was uttered by the same individual for all three analysed species. From a first qualitative observation of our datasets, fish sound sequences appeared highly variable between taxa. Therefore, rhythm analysis was conducted over an acoustic sample of a specific duration, which was defined *a priori* to be tailored on the specific sequences' characteristics of each sound type. The acoustic samples of *O. rochei* were 10 min long ($N = 10$), while those of *S. umbra* ($N = 10$) and /Kwa/ ($N = 20$) were 1 min long. In the case of *S. umbra*, the 1 min acoustic sample contained either a sequence or a portion of it; in the case of the /Kwa/ sound type, the entire sequence was always fully included in the chosen acoustic sample. As we never had more than one sequence per sample, no other threshold/boundary to determine sequences was needed for the data at hand. On average, an individual acoustic sample contained $28 \pm 7$; $10 \pm 4$ and $5 \pm 1$ (mean ± s.d.) elements of the *S. umbra*, /Kwa/ and *O. rochei* sound type, respectively.

The acoustic samples were analysed by audio and visual assessment using Raven Pro 64 1.4 (Bioacoustic Research Program, Cornell Laboratory of Ornithology, Ithaca, NY, USA; sound files downsampled to 4 kHz, fast Fourier transform (FFT) size 256 points, 50% overlap, Hanning window). Each element was selected in the oscillogram from its onset to its offset using the selection function in Raven, which generates a selection table. In the Raven selection table, each selection made by the operator (i.e. each element) is labelled by its start and end times. The Raven selection tables were then exported to Excel to calculate the IOIs and subsequent analyses.

## 2.2. Assessment of rhythm parameters

For rhythm analysis, we followed the workflow developed by Burchardt & Knörnschild [2], to which we refer the reader for specific details for example on equations. For each sound type, a visual inspection of

the distribution of IOIs was carried out, by creating a histogram. Bin width for the histograms was calculated following Sturges' rule [34]. The shape of the distribution gives a first impression on whether one can assume an underlying isochronous patterning. For example, a unimodal distribution suggests isochrony; a bimodal distribution hints at periodicity but not necessarily isochrony and finally, a uniform distribution might hint at no periodicity or rhythm. Furthermore, the **normalized pairwise Variability Index (nPVI)** values and **coefficients of variation ($C_V$)** were calculated for each sequence: nPVI is a measure of the average variation of a set of durations (IOIs), obtained from successive ordered pairs of elements, which describes how well the next IOI can be predicted based on the previous IOI. The nPVI would be 0 in a perfectly isochronous sequence. Higher nPVI values correspond to a higher variability between IOIs. While a low nPVI value (less than 30) is a strong indicator for an underlying isochronous pattern, high values do not exclude an isochronous pattern, especially in the case of long sequences [15,35]. The $C_V$ value is the standard deviation set into relation with the sample mean, giving a value which is independent of sample size and mean, and therefore providing a comparable description of a sequence's variability: the smaller it is, the lower the variability and therefore the higher the probability of isochrony. For both values, no clear thresholds exist yet on which values indicate isochrony or rhythmicity, and which do not.

Recurrence plots provided a visual representation of the sequence's temporal pattern. Depicted are squares, each square represents one comparison of IOIs; for example, the first square on the bottom left is the comparison between the first IOI with the first IOI, the following one is the comparison between the first IOI with the second IOI, and so on. The plot is mirrored on the diagonal. Every square has the same size, and the number of squares depicts the number of IOIs in the sequence. The number of squares (i.e. IOIs) plus one is the number of elements in the whole sequence. A Euclidean distance is calculated to quantify the difference between two IOIs, the distance is then colour-coded in the plot. The colour-coding ranges from white (i.e. no difference between IOIs) to black (i.e. the maximum difference between IOIs in this dataset). Depending on the actual differences depicted in the plot, white will always mean 'no difference' while black can stand for very differently sized distances. Colour-coding is sequence-specific since standardizing for a dataset comprising large distances would result in significantly losing resolution for smaller distances. Recurrence plots were realized with Matlab 2017b; all other visualizations were realized in R Studio (v. 1.3.959, R v. 4.0.1).

Finally, a best-fitting exact **beat frequency** was calculated for each sequence. The beat describes the underlying isochronous pattern of a sequence. It is measured as elements per second, and it is, therefore, a frequency (in Hz). For example, a 5 Hz beat would describe a sequence with a beat every 200 ms, corresponding to the element onsets; however, silent beats (i.e. a beat not accompanied by an element) might occur. In this context, we must make the distinction between so-called 'signal isochrony' and 'induced isochrony'. On one hand, we can imagine a sequence where the signal itself is isochronous, like for example whale echolocation sequences [2]. That would be called 'signal isochrony'. On the other hand, we could have sequences that are well described by an underlying isochronous beat, but there are 'silent beats' of the theoretical underlying beat that are not accompanied by an element in the element sequence. The perception of isochrony might be induced even though it is not there. This so-called 'beat induction' is argued to be a fundamental musical trait in humans [36,37] and an important notion to keep in mind when analysing animals' acoustic signals ('induced isochrony' [38]). This is especially true, as for many animals we do not know how they perceive the rhythmic signals. The distinction is still valid for descriptive purposes.

We calculated best-fitting beats in two ways: (i) using the sequence of IOIs and averaging them to calculate how many beats are found, on average, per second (Hz) and (ii) by calculating a fast Fourier transformation in a frequency window of 1 to 10 Hz. Since up to half of a given sampling rate can be decomposed by a Fourier analysis, original data were transformed into a time sequence with a 20 Hz sampling rate. The custom-made Matlab codes for the Fourier analysis, also including IOI calculations, as well as calculations of nPVI and $C_V$, can be found at: https://github.com/LSBurchardt.

# 3. Results and interpretation

## 3.1. Regular temporal structure of elements: brown meagre (*Sciaena umbra*) sounds

The brown meagre shows an isochronous rhythm. This was qualitatively suggested by visual inspection of the sequence (figure 1*a*) and was confirmed by the unimodal distribution of the IOIs (figure 2*a*), which indicates similar IOI duration and, therefore, signal isochrony. The exemplary recurrence plot shows no

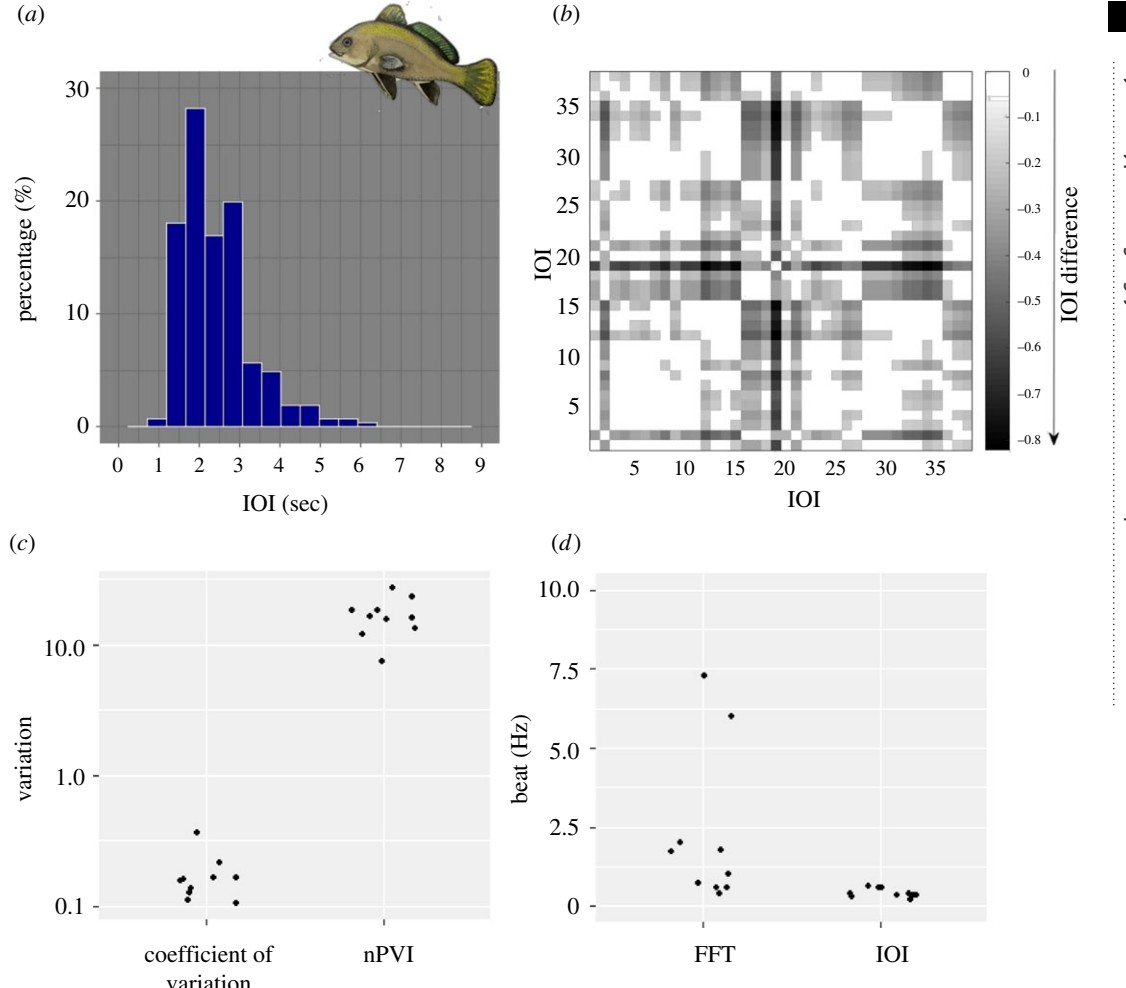

**Figure 2.** Rhythm analysis on sounds produced by the brown meagre. (*a*) Histogram of the duration of IOIs for all sequences analysed for the brown meagre. Category width as calculated by Sturges' rule and lies at approximately 0.5 s, y-axis as percentages. We see a unimodal distribution hinting at an isochronous rhythm. (*b*) Recurrence plot of a typical sound sequence. Note in the colour-coding, that differences are small, no substructure (i.e. clear squares or patterns) can be seen. (*c*) Variation parameters: coefficient of variation and nPVI for the 10 sound sequences of the brown meagre, *Sciaena umbra*. Plotted on a log-scale for y, both parameters are small and support underlying isochrony. (*d*) Best-fitting beats analysed for the 10 example sequences with the Fourier analysis (FFT) and beat calculations based on IOIs show beats between 0 and 7.5 Hz.

subpatterns, but a consistent structure of IOIs with only small, irregular differences (figure 2*b*). Furthermore, the brown meagre sequence was characterized by low nPVI values (maximum = 27.36, median = 16.07), as well as by low $C_V$ values (maximum = 0.36, median = 0.16) (figure 2*c*). The low nPVI and $C_V$ values indicate a low variability between IOI duration; in other words, elements are evenly distributed in time and, therefore, follow an isochronous pattern. The best-fitting beats of the brown meagre's sequences were calculated at frequencies between 0.22 and 0.62 Hz (median = 0.39 Hz) following the IOI analysis, and 0.42 and 7.3 Hz (median = 1.37 Hz) following the Fourier analysis (figure 2*d*). Two clusters are visible in figure 2*d*: one between 0 and 2.5 Hz and one with two faster beats around 7 Hz, which are also highlighted by the Fourier analysis. Nevertheless, the faster beat is not evident when using the IOI analysis. Looking only at the 10 analysed sequences, it seems that intra-specific variability with regard to beat is low.

## 3.2. Unclear temporal structure of elements: /Kwa/ sounds

Sequences of /Kwa/ also show an isochronous pattern, but not as clearly as in the case of the brown meagre. In contrast with the sequences of the brown meagre, we see different IOI duration: very short IOIs and others that are longer in duration. This generates a bimodal distribution of IOIs in the histogram (figure 3*a*) supporting a clear alternation of two types of IOIs that can be observed in the exemplary recurrence plots (figure 3*b*). Since the most prominent categories of IOI durations are

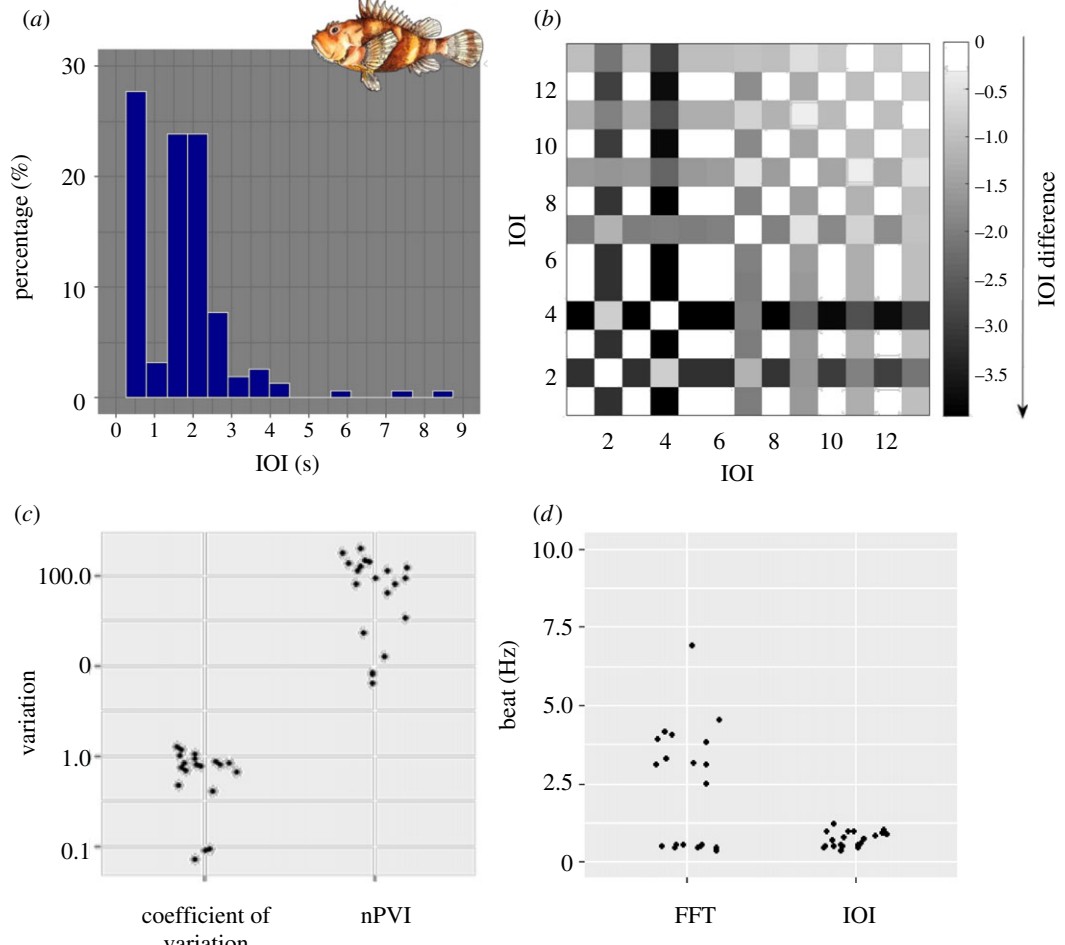

**Figure 3.** Rhythm analysis on the /Kwa/ sound type produced by *Scorpaena spp*. (*a*) Histogram of the duration of IOIs for all sequences analysed for the /Kwa/ sound. Category width as calculated by Sturges' rule and lies at approximately 0.5 s, y-axis as percentages. We see a bimodal distribution, hinting at periodicity but not necessarily isochrony. (*b*) Recurrence plot of a typical sound sequence. Note in the colour-coding, that differences are bigger than for the brown meagre but still small, a clear alternation of IOIs can be seen, describing a sequence, where short and longer IOIs take turns. (*c*) Variation parameters: coefficient of variation and nPVI for the 20 /Kwa/ sounds. Plotted on a log-scale for y, both parameters show a considerable variation of IOI durations. Isochrony cannot be expected based on these parameters alone. (*d*) Best-fitting beats analysed for the 20 example sequences with the Fourier analysis (FFT) and beat calculations based on IOIs show beats between 0 and 7.5 Hz.

multiples of each other (lying at 0.5 and 2 s), the sequences of /Kwa/ still present an underlying isochronous rhythm. However, in contrast with *S. umbra* sequences, this isochronous rhythm is characterized by both beats fitting well to vocal elements and beats that do not correspond to any element (i.e. silent beats). We, therefore, speak of 'induced isochrony' for the /Kwa/ sequences.

The nPVI and $C_V$ values mirror the variability in IOI durations (figure 3*c*). We find low nPVI values, indicating that we do have sequences that consist of evenly spaced IOIs, but also very high nPVIs, indicating sequences with variable IOI durations. The same is true for $C_V$ values (nPVI: 6.29 to 197.11, median = 90.9 and $C_V$: 0.07 to 1.23, median = 0.79) (figure 3*c*). The higher $C_V$ and nPVI, the more variable are IOI durations in the sequence.

The best-fitting beats range from 0.33 to 1.19 Hz following the IOI analysis (median = 0.7) and from 0.33 to 6.88 Hz following the Fourier analysis (median = 2.79); again, we see two groups of beats (figure 3*d*).

## 3.3. Random temporal structure of elements: *Ophidion rochei* sounds

Element sequences of *Ophidion rochei* do not show any clear temporal patterning (figure 1*c*). This is confirmed by the relatively uniform distribution of IOIs, as well as by the large range of durations

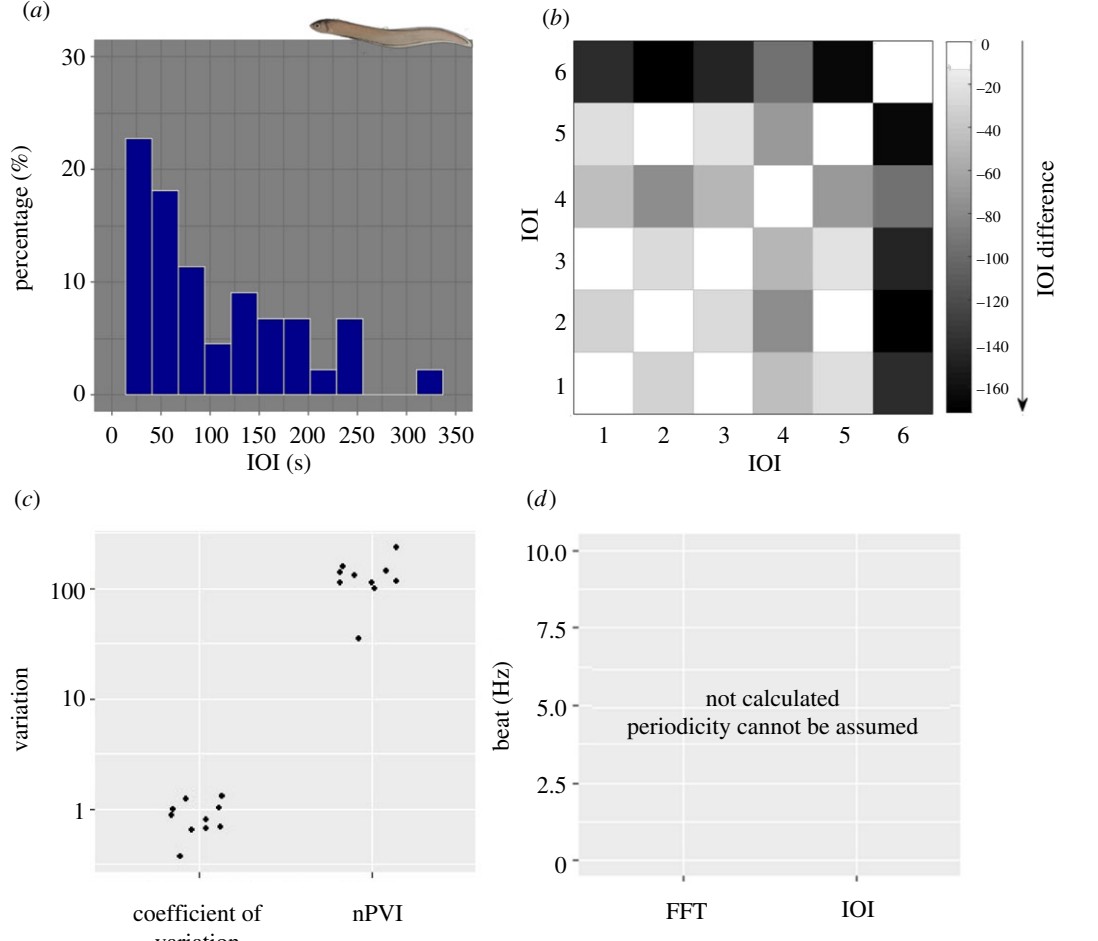

**Figure 4.** Rhythm analysis for sounds produced by *Ophidion rochei*. (*a*) Histogram of the duration of IOIs for all sequences analysed for *O. rochei* sounds. Category width as calculated by Sturges' rule and lies at approximately 26 s, y-axis as percentages. We see a unimodal distribution, the large bin width and variation of IOI durations hints at a random patterning. (*b*) Recurrence plot of a typical sound sequence. Note in the colour-coding, that differences are 50- to 160-fold larger than for the other examples. (*c*) Variation parameters: coefficient of variation and nPVI for the 10 *O. rochei* sounds. Plotted on a log-scale for y, both parameters show a high variation of IOI durations. Isochrony cannot be expected. (*d*) Best-fitting beats were not calculated, as we cannot assume an underlying rhythmic structure.

visible on the x-axis of figure 4*a* compared with the other species (see figures 2*a* and 3*a* versus 4*a*). In fact, even if one category appears as prominent because it includes approximately 25% of IOIs, this has a very broad range of variation (i.e. 26 s, figure 4*a*) compared with the other species; the difference is striking (approx. 0.5 s, figures 2*a* and 3*a*). The exemplary recurrence plot shows no clear structure, and the first visual impression of small differences between IOIs (figure 4*b*) is corrected when considering the distance values are 50- to 160-fold higher than in the other two examples (figures 2*b* and 3*b*).

The nPVI values are very high on average (nPVI: 35.53 to 235.8, median = 124.14; figure 4*c*), further confirming a random pattern in the element sequence, with only two out of 10 sequences having nPVI values below 100. The $C_V$ values lie between 0.65 and 1.32 with a median of 0.83; these values are not very different from the results we got for the /Kwa/ sounds, which shows that one parameter alone is inconclusive. Due to the results of the preceding analyses, we decided to not calculate best-fitting beats, as these would be inconclusive because we cannot assume an underlying isochronous pattern.

## 4. Conclusion and perspective

The rhythms of animals' acoustic signals, produced internally or by tool-use, were so far quantified in birds [13,39], sea mammals [2,7,40], bats [2,12], primates [41] and, to some extent, also in insects [42], but they were never quantitively measured in fish. This manuscript set out to fill this gap by

providing the first quantification of fish sounds rhythms and more importantly, by testing this methodology on fish sound sequences characterized by different rhythmical properties. The wide range of variability presented here should aid future applications and replication, at both the analysis and the results interpretation level. The methodological approach presented here demonstrated that it is possible to mathematically quantify the periodicity of fish sound sequences recorded at sea or, in other words, that it is possible to define fish vocal rhythms in natural environments. Furthermore, this paper confirms the initial, qualitative assessment of the rhythmical properties of the three most studied sonic fish taxa in the Mediterranean Sea: (i) simple and isochronous (*Sciaena umbra*), (ii) more complex with underlying isochrony (*Scorpaena* spp.), and (iii) no rhythmic structure (*Ophidion rochei*). Here are some key features that should be considered in the use of the methodology.

Looking at general prerequisites to execute the proposed rhythm analysis (especially the Fourier analysis to calculate exact beat frequencies for single sequences) an important factor to be considered is sequence duration. In fact, sequences shorter than approximately 1 s would result in a very low frequency resolution in the Fourier analysis (as was described for the bat *Carollia perspicillata* in [2]), which makes the results potentially imprecise. Sequence durations of the here-analysed fish sound sequences were always longer than 1 s, making them good candidates for the here-proposed methodology.

The analysed brown meagre sounds show a strong isochronous rhythm, in accordance with the qualitative assessment of the R-pattern in previous studies [25,33]. The low nPVI and CV values are comparable to the values found for echolocation sequences of the sperm whale (*Physeter macrocephalus*), which have a very consistent isochronous structure [2]; the two sound types show the same kind of isochronous rhythm, merely with different beats. We can speak of signal isochrony. The intra-species variability between the 10 considered sequences is low, and *S. umbra* calls seem to be uttered with similar beats independently of the emitting individual. Nevertheless, there might be at least two different beats present in this species, when looking at the Fourier analysis results. By expanding the dataset, the here-proposed methodology could help in further evaluating context-related beat variability, as well as in exploring the existence of individual-related beats within the isochronous rhythm emitted by this species.

The /Kwa/ sound sequences analysed here were characterized by more complex rhythmical properties than those characterizing brown meagre sounds. However, these could still correspond to an isochronous pattern, including silent beats and therefore 'induced isochrony'. The $C_V$ values are much higher than reported for clearly isochronous sequences of bats and whales [2]. Indeed, these higher values are due to the variability of the IOIs within a sequence: /Kwa/ sequences are characterized by alternating IOI successions of short and long IOIs; this rhythm is, therefore, less intuitive to be qualitatively described. Nevertheless, both IOI types fit well to the calculated beats, as the IOI types (short and long durations) are multiples of each other. This leads to an increase in variability but not necessarily to a decrease in how well a calculated beat describes a sequence. There are no clear thresholds for $C_V$ or nPVI values, stating which values indicate isochrony or rhythmicity and which do not. Studies such as ours help in the interpretation and evaluation of calculated values. Values for artificial sequences were also published and are another helpful addition [15]. In the present dataset, the methodology highlighted two classes of beats found in different sequences. Further applications of this methodology could involve testing the hypothesis that beats might play a role in species-specific discrimination within this species complex. The /Kwa/ sound type, which dominates the soundscape of *Posidonia oceanica* meadows, is indeed probably emitted by *Scorpaena* spp., and it is characterized by wide sound features variability (e.g. element duration, figure 1*b*). At the current state of knowledge, /Kwa/ species-specific acoustic features have not yet been identified. In a study carried out over large spatial scales and on extremely large sample sizes (greater than 10 k sounds), principal component analysis on sound features revealed more clusters than the number of *Scorpaena* species present on site [43]; this, in its turn, raised the question of whether species-specific information might be encoded using other kinds of features, for example, in the intervals between elements. Further studies should shed light on the hypothesis that different *Scorpaena* species might code specific identity through beats and that indeed the twobeat classes found could belong to different *Scorpaena* species.

*Ophidion rochei* emits sounds without a consistent temporal structure. By contrast, an evident temporal structure can be found within one element [44]. We chose this species to show a non-rhythmic example and what kind of elements are not suitable for rhythm analysis.

The present paper shows exemplary results of rhythm analysis for three different kinds of temporal structures in fish sounds. In general, it might be advantageous to produce rhythmic acoustic signals because 'rhythmic attention' (*sensu* [45]) helps receivers to decode rhythmic signals easier and

faster [46]. When rhythms exist, the receiver's attention cycles in an oscillatory way [47,48]. Since rhythmic signals are predictable, 'rhythmic attention' enables receivers to provide most 'attentional energy' at a time point where the next stimulus is to be expected. This is advantageous because cognitive capacities are limited [49] and optimization of attention timing is helpful to not miss relevant stimuli. An example from the gregarious Old World monkeys macaques (*Macaca fascicularis*) shows that neuronal oscillations in the primary visual cortex entrain to a stimuli stream (visual stimuli) when the stream is rhythmic, a mechanism resulting in decreased reaction time and an increase in the response gain for events that are task-relevant [50]. Correspondingly, rhythmicity in fish acoustic signals might be adaptive for saving metabolic energy in a potentially noisy environment, where signaller and receiver most often do not see each other. Attentional tuning might be very important if communication occurs over long distances. This hypothesis is of course only valid in the case that fish can perceive the rhythmic structure, which is very likely but has not yet been shown.

From the methodological point of view, this manuscript has demonstrated that the presented methods are effective to quantify rhythms of fish acoustic signals. In contrast with previous studies, we only tested for beat frequencies of up to 10 Hz, because of the lower number of elements per second in comparison with, for example, bats, whose acoustic signals show exact best-fitting beat frequencies of up to 100 Hz [2,12]. Best-fitting beats for the analysed fish sounds matched well between methods, further proving that these two methods—Fourier analysis and IOI rhythm calculations—are suitable for analysing this kind of data. These methodologies are highly recommended for the analysis of fish sound rhythms and a wide range of applications can be suggested. Among others, rhythm analysis could be used to distinguish species, behavioural contexts as well as individual identity. Furthermore, it would be worthwhile to investigate geographical differences associated with beats, which potentially inform about the relationship between beats and local soundscape features. These considerations are of interest to different fields, spanning from ecology, ethology and conservation biology to neurobiology. Understanding the underpinnings of such beats, and how they are plastically produced is a fundamental, unanswered question in fish bioacoustics. We hope this manuscript encourages the use of these methodologies for rhythm analysis in fish for a yet larger set of questions and enables more researchers to follow up on them.

Ethics. No permits were needed for conducting this research. Recordings were obtained by using the strictly observational and non-invasive technique of passive acoustic monitoring (PAM). PAM involves the deployment of acoustic data loggers in the marine environment; no animals nor animal samples were collected, and no animal procedure was carried out.

Data accessibility. The raw sequence and IOI data is available at the MfN Data Repository at https://doi.org/10.7479/6qfa-9956. Code for the rhythm analysis is available at https://github.com/LSBurchardt/FFT-Method.

Authors' contribution. M.B. and L.S.B. conceived the idea. M.B. and M.P. collected and prepared the acoustic data. L.S.B. and M.B. analysed the data following the computational framework developed by L.S.B. E.P. provided part of the financial resources and recording equipment. L.S.B. wrote the first manuscript draft. All authors discussed the results and contributed to the final manuscript.

Competing interests. We declare we have no competing interests.

Funding. M.B. was funded by a BeIPD-Marie Curie COFUND postdoctoral fellowship at the University of Liège (Belgium) and by an Opportunity grant issued by University of Liège (DORIS, contract no. 09/2020). L.S.B. was funded by an Elsa-Neumann Scholarship of the Landesgraduiertenförderung Berlin, Germany. The publication of this article was funded by the Open Access Fund of the Leibniz Association.

Acknowledgements. We would like to thank Ignacio A. Catalan and Josep Alós for field support in Mallorca, as well as A. Codarin, S. Malavasi and C. Facca for supporting data acquisition in Trieste and Venice (Italy). We thank two anonymous reviewers for their helpful comments and Dr Anthony D. Hawkins for his valuable feedback and proof reading.

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
