## [Peer Review File · Royal Society Open Science]

Review History

RSOS-210494.R0 (Original submission)

Review form: Reviewer 1

Is the manuscript scientifically sound in its present form?

Yes

Are the interpretations and conclusions justified by the results?

Yes

Is the language acceptable?

Yes

Do you have any ethical concerns with this paper?

No

Have you any concerns about statistical analyses in this paper?

No

Recommendation?

Accept with minor revision (please list in comments)

Comments to the Author(s)

Thank you for the opportunity to review this paper. I enjoyed reading it and believe this work will be of interest to the broader marine bioacoustics community, particularly fish biologists. In the paper, Burchardt et al. use a variety of methods to quantify rhythmic structure in several fish sounds. I think the framing of the article in the introduction and conclusion was very well done. It was clear what research gap was being filled (e.g. lines 67-70) and how the methodology/results could be extended to future work (various portions of the discussion). I also appreciated that the authors chose to highlight three taxa with different qualitative interpretations of inherent rhythmicity, as it gives readers who are less familiar with rhythmic analysis a first look at how different rhythms (or the absence of rhythm) can manifest in acoustic signals. The methodology was generally quite clear. For example, the authors included a very clear but concise description of recurrence plots (lines 170-181). All four figures were excellent.

Major comments

I was confused by the authors' interpretation of the /Kwa/ sound type results, because many of their results and discussion points (e.g. the bimodal IOI distribution, the "clear alternation of two types of IOIs that can be observed in the recurrence plots") suggested to me that the /Kwa/ sound type might be heterochronous, not isochronous. My question for the authors is this: of the two beats that they found for the /Kwa/ sound type, do they know that each sequence only ever had one of those two beats (in which case I would agree that the sequences are isochronous)? Or do single sequences contain predictable alterations of short and long intervals (in which case it seems that there is some evidence for heterochrony)? I wonder if it is the latter since the authors write in the Figure 2 caption that they "see a bimodal distribution, hinting at periodicity but not necessarily isochrony." I think it is really important to clarify this point in the manuscript.

I was also confused by how exactly the authors determined boundaries between sequences. They discuss a "prolonged silence" in line 64, then say in line 98 that "a sequence is terminated after a defined (i.e. 30 seconds) duration of silence." Looking at figure 1C, however, the silence between the first two elements and the third one is much longer than 30 seconds, but those elements are labeled as belonging to the same sequence. This makes me think that the authors did not use 30 seconds as their benchmark. An explicit discussion of how sequence boundaries were non-arbitrarily determined for the three taxa would be helpful. I believe the authors start to get at this in lines 134-137, but I found those lines (particularly 136-137) confusing.

Minor comments

The paper would benefit from a good round of editing to remove some small (but sometimes distracting) typographic issues. There are several run-on sentences (e.g. lines 170-171, sentence starting with "Depicted..."; lines 44-46, sentence starting with "Important..."; portion of the Figure 2/3 captions that start with "Note in the color-coding..." etc.). The authors say that terms from Table 1 are italicized in the text, but many additional words are italicized as well (e.g. periodicity within, within, within-sound, between; see 'Background' paragraph 2). I found this confusing. There seems to be missing punctuation in the captions for Figures 2/3/4 (lines starting with "variation parameters coefficient...").

A few other minor things:

- Bit rate should be given for all recording devices (missing for line 124). If that information is in the cited paper (reference #24), make that clearer.
 - GPS presentation should be consistent (e.g. line 127 format vs. line 131).
 - Put earlier (than lines 232-233) that each recurrence plot in Figures 2/3/4 are exemplary.
- It would be great to have the recurrence plots for each sequence analyzed as supplement. I know this is more work, but the authors do provide the C_v , nPVI, FFT, and IOI for all sequences; the one thing that is not currently available is the recurrence plots.

- Put lines 282-283 (about lack of Cv and nPVI thresholds) sooner in the paper.
- I could download the files of raw data from the MfN data repository and open them using Notepad, but they seem to be semicolon separated files instead of comma separated files. I tested this on two different PC computers and got the same results.

I also have a few lingering questions, which do not necessarily need to be addressed in the paper, but might come up for other readers as well:

- Why did you choose a 1-minute duration for the *S. umbrae* acoustic samples when that did not always include the full sequence? It sounds like you set the /Kwa/ duration to always include entire sequences but did not do the same for *S. umbrae*.
- Did you do signal-to-noise ratio analyses to ensure that the sequences you recorded were from single individuals? You mention that SNR can be used to do this (lines 137-139), but do not say if you did it (although lines 269-271 suggest you might have).

Review form: Reviewer 2

Is the manuscript scientifically sound in its present form?

Yes

Are the interpretations and conclusions justified by the results?

Yes

Is the language acceptable?

Yes

Do you have any ethical concerns with this paper?

No

Have you any concerns about statistical analyses in this paper?

No

Recommendation?

Accept with minor revision (please list in comments)

Comments to the Author(s)

Burchardt et al. recorded sound from three marine fish species from different genera and described and analyzed rhythmicity in them. They used pure descriptive, mathematical, and descriptive statistical methods to assess the rhythm in these sounds. They describe all technical acoustic terminology precisely and make their research accessible to a wide audience in biology and hence to researchers interested in using these results for potentially interdisciplinary purposes.

The paper is well written and clear, the illustrations are accessible and the findings a valuable contribution. I strongly recommend to accept this work for publication in RSOS.

I have only a few comments and suggestions; please see below. One minor issue is that the authors appear to imply that fish and other species can perceive rhythm as such. The perception of patterns is most probably wide spread in animals and I have no doubt that fish are capable of it. But, to my knowledge, it has not been empirically demonstrated and this still needs to be done. It is well possible that fish use rhythmicity to discriminate between species and it is fully justified

to discuss this possibility; it might be worth mentioning though that this hypothesis requires also studies on perception; and particularly the perception of rhythm and not just natural calls.

Comments

Line 23: Is this "C." standing for something?

Line 36 Rhythm as such is, to my knowledge, not coding for urgency in meerkats. The calls are categorically distinct. That is, the calls as a holistic unit are perceived as more urgent.

Line 39-40: This only applies if the animals are sensitive to sequential dependencies, which is not shown in many species.

Line 50: Please, merge references.

Line 184: Would that not be every 200 ms?

Line 238-241: This is excellent, clear and considered.

Line 310-312 (and to a degree above): I in principle agree with all of that. As far as I know though, it has not yet been shown whether fish are sensitive to rhythm in acoustic sequences; that is: perceiving the rhythm of the sequence itself and not the just a sound with a given rhythm. I would be highly surprised if they could not, but as long as it is not shown, there is uncertainty.

Decision letter (RSOS-210494.R0)

Dear Ms Burchardt

On behalf of the Editors, we are pleased to inform you that your Manuscript RSOS-210494 "A primer on rhythm quantification for fish sounds: a Mediterranean case study" has been accepted for publication in Royal Society Open Science subject to minor revision in accordance with the referees' reports. Please find the referees' comments along with any feedback from the Editors below my signature.

Please submit your revised manuscript and required files (see below) no later than 7 days from today's (ie 21-Jul-2021) date. Note: the ScholarOne system will 'lock' if submission of the revision is attempted 7 or more days after the deadline. If you do not think you will be able to meet this deadline please contact the editorial office immediately.

Please note article processing charges apply to papers accepted for publication in Royal Society Open Science (<https://royalsocietypublishing.org/rsos/charges>). Charges will also apply to papers transferred to the journal from other Royal Society Publishing journals, as well as papers

submitted as part of our collaboration with the Royal Society of Chemistry (<https://royalsocietypublishing.org/rsos/chemistry>). Fee waivers are available but must be requested when you submit your revision (<https://royalsocietypublishing.org/rsos/waivers>).

on behalf of Pete Smith (Subject Editor)
openscience@royalsociety.org

Associate Editor Comments to Author:

Comments to the Author:

Please ensure that your revision clearly indicates the changes made in a tracked-changes version (as well as providing a 'clean' version), and a full, point-by-point response to the reviewers' comments.

Reviewer comments to Author:

Reviewer: 1

Comments to the Author(s)

Thank you for the opportunity to review this paper. I enjoyed reading it and believe this work will be of interest to the broader marine bioacoustics community, particularly fish biologists. In the paper, Burchardt et al. use a variety of methods to quantify rhythmic structure in several fish sounds. I think the framing of the article in the introduction and conclusion was very well done. It was clear what research gap was being filled (e.g. lines 67-70) and how the methodology/results could be extended to future work (various portions of the discussion). I also appreciated that the authors chose to highlight three taxa with different qualitative interpretations of inherent rhythmicity, as it gives readers who are less familiar with rhythmic analysis a first look at how different rhythms (or the absence of rhythm) can manifest in acoustic signals. The methodology was generally quite clear. For example, the authors included a very clear but concise description of recurrence plots (lines 170-181). All four figures were excellent.

Major comments

I was confused by the authors' interpretation of the /Kwa/ sound type results, because many of their results and discussion points (e.g. the bimodal IOI distribution, the "clear alternation of two types of IOIs that can be observed in the recurrence plots") suggested to me that the /Kwa/ sound type might be heterochronous, not isochronous. My question for the authors is this: of the two beats that they found for the /Kwa/ sound type, do they know that each sequence only ever had one of those two beats (in which case I would agree that the sequences are isochronous)? Or do single sequences contain predictable alterations of short and long intervals (in which case it seems that there is some evidence for heterochrony)? I wonder if it is the latter since the authors write in the Figure 2 caption that they "see a bimodal distribution, hinting at periodicity but not necessarily isochrony." I think it is really important to clarify this point in the manuscript.

I was also confused by how exactly the authors determined boundaries between sequences. They discuss a "prolonged silence" in line 64, then say in line 98 that "a sequence is terminated after a defined (i.e. 30 seconds) duration of silence." Looking at figure 1C, however, the silence between

the first two elements and the third one is much longer than 30 seconds, but those elements are labeled as belonging to the same sequence. This makes me think that the authors did not use 30 seconds as their benchmark. An explicit discussion of how sequence boundaries were non-arbitrarily determined for the three taxa would be helpful. I believe the authors start to get at this in lines 134-137, but I found those lines (particularly 136-137) confusing.

Minor comments

The paper would benefit from a good round of editing to remove some small (but sometimes distracting) typographic issues. There are several run-on sentences (e.g. lines 170-171, sentence starting with "Depicted..."; lines 44-46, sentence starting with "Important..."; portion of the Figure 2/3 captions that start with "Note in the color-coding..." etc.). The authors say that terms from Table 1 are italicized in the text, but many additional words are italicized as well (e.g. periodicity within, within, within-sound, between; see 'Background' paragraph 2). I found this confusing. There seems to be missing punctuation in the captions for Figures 2/3/4 (lines starting with "variation parameters coefficient...").

A few other minor things:

- Bit rate should be given for all recording devices (missing for line 124). If that information is in the cited paper (reference #24), make that clearer.
- GPS presentation should be consistent (e.g. line 127 format vs. line 131).
- Put earlier (than lines 232-233) that each recurrence plot in Figures 2/3/4 are exemplary. It would be great to have the recurrence plots for each sequence analyzed as supplement. I know this is more work, but the authors do provide the Cv, nPVI, FFT, and IOI for all sequences; the one thing that is not currently available is the recurrence plots.
- Put lines 282-283 (about lack of Cv and nPVI thresholds) sooner in the paper.
- I could download the files of raw data from the MfN data repository and open them using Notepad, but they seem to be semicolon separated files instead of comma separated files. I tested this on two different PC computers and got the same results.

I also have a few lingering questions, which do not necessarily need to be addressed in the paper, but might come up for other readers as well:

- Why did you choose a 1-minute duration for the *S. umbrae* acoustic samples when that did not always include the full sequence? It sounds like you set the /Kwa/ duration to always include entire sequences but did not do the same for *S. umbrae*.
- Did you do signal-to-noise ratio analyses to ensure that the sequences you recorded were from single individuals? You mention that SNR can be used to do this (lines 137-139), but do not say if you did it (although lines 269-271 suggest you might have).

Reviewer: 2

Comments to the Author(s)

Burchardt et al. recorded sound from three marine fish species from different genera and described and analyzed rhythmicity in them. They used pure descriptive, mathematical, and descriptive statistical methods to assess the rhythm in these sounds. They describe all technical acoustic terminology precisely and make their research accessible to a wide audience in biology and hence to researchers interested in using these results for potentially interdisciplinary purposes.

The paper is well written and clear, the illustrations are accessible and the findings a valuable contribution. I strongly recommend to accept this work for publication in RSOS.

I have only a few comments and suggestions; please see below. One minor issue is that the authors appear to imply that fish and other species can perceive rhythm as such. The perception of patterns is most probably wide spread in animals and I have no doubt that fish are capable of

it. But, to my knowledge, it has not been empirically demonstrated and this still needs to be done. It is well possible that fish use rhythmicity to discriminate between species and it is fully justified to discuss this possibility; it might be worth mentioning though that this hypothesis requires also studies on perception; and particularly the perception of rhythm and not just natural calls.

Comments

Line 23: Is this "C." standing for something?

Line 36 Rhythm as such is, to my knowledge, not coding for urgency in meerkats. The calls are categorically distinct. That is, the calls as a holistic unit are perceived as more urgent.

Line 39-40: This only applies if the animals are sensitive to sequential dependencies, which is not shown in many species.

Line 50: Please, merge references.

Line 184: Would that not be every 200 ms?

Line 238-241: This is excellent, clear and considered.

Line 310-312 (and to a degree above): I in principle agree with all of that. As far as I know though, it has not yet been shown whether fish are sensitive to rhythm in acoustic sequences; that is: perceiving the rhythm of the sequence itself and not the just a sound with a given rhythm. I would be highly surprised if they could not, but as long as it is not shown, there is uncertainty.

===PREPARING YOUR MANUSCRIPT===

If you have been asked to revise the written English in your submission as a condition of publication, you must do so, and you are expected to provide evidence that you have received language editing support. The journal would prefer that you use a professional language editing service and provide a certificate of editing, but a signed letter from a colleague who is a native

speaker of English is acceptable. Note the journal has arranged a number of discounts for authors using professional language editing services (<https://royalsociety.org/journals/authors/benefits/language-editing/>).

===PREPARING YOUR REVISION IN SCHOLARONE===

-- If you have uploaded ESM files, please ensure you follow the guidance at <https://royalsociety.org/journals/authors/author-guidelines/#supplementary-material> to include a suitable title and informative caption. An example of appropriate titling and captioning

may be found at https://figshare.com/articles/Table_S2_from_Is_there_a_trade-off_between_peak_performance_and_performance_breadth_across_temperatures_for_aerobic_sc_ope_in_teleost_fishes_/3843624.

Author's Response to Decision Letter for (RSOS-210494.R0)

See Appendix A.

Decision letter (RSOS-210494.R1)

Dear Ms Burchardt,

I am pleased to inform you that your manuscript entitled "A primer on rhythm quantification for fish sounds: a Mediterranean case study" is now accepted for publication in Royal Society Open Science.

Kind regards,
Royal Society Open Science Editorial Office

on behalf of Pete Smith (Subject Editor)
openscience@royalsociety.org

Appendix A

26.07.2021

To whom it may concern,

I am Professor Anthony Hawkins CBE FRSE, the former Head of Fisheries Research for Scotland. I hereby confirm that I have read and edited the manuscript "A primer on rhythm quantification for fish sounds: a Mediterranean case study." by Lara S. Burchardt, Marta Picciulin, Eric Parmentier and Marta Bolgan. Much of the text is very well written, but I have suggested a few small changes to improve the English language that is being used.

Best regards,

Professor Anthony Hawkins CBE FRSE

Aberdeen, Scotland